civil engineering/materials science/
structural engineering

water absorption, cement-based materials,
X-ray CT, durability

**Author for correspondence:**
Danying Gao
e-mail: gdy@zzu.edu.cn

# Relationship between sorptivity and capillary coefficient for water absorption of cement-based materials: theory analysis and experiment

Lin Yang[1], Danying Gao[1], Yunsheng Zhang[3], Jiyu Tang[2] and Ying Li[4]

[1]School of Water Conservancy and Environment, and [2]School of Civil Engineering, Zhengzhou University, Zhengzhou 450001, People's Republic of China
[3]School of Materials Science and Engineering, Southeast University, Nanjing 211189, People's Republic of China
[4]Henan Key Laboratory of Intelligent Manufacturing of Mechanical Equipment, Zhengzhou University of Light Industry, Zhengzhou 450002, People's Republic of China

LY, 0000-0002-6910-2596

The durability of cement-based materials depends on the property of water absorption. In this work, a technique of X-ray CT combined with CsCl enhancing was used to continuously monitor the dynamic process of water uptake in cement-based materials and the gravimetric method was used to measure the amount of water absorption. The relationship between the capillary coefficient ($k$) and sorptivity ($S$) was firstly established based on theory analysis and well verified by the experiment results. In accordance with theory analysis and experiment results, it is found that the ratio of sorptivity to capillary coefficient equals the porosity ($\varphi$) of materials, i.e. $S/k = \varphi$. This model provides a simple method for obtaining the capillary coefficient of porous materials from the measurement of sorptivity and porosity.

## 1. Introduction

Cement-based materials are widely used in civil engineering, traffic engineering, ocean engineering, water management, national defence, etc., and play an important role in the national infrastructure construction [1–3]. In people's traditional views, cement-based materials have excellent durability. However, many

concrete structures begin to deteriorate when they are in service for several years, owing to factors such as chloride attack, sulfate attack, frost, carbonation, alkali-silica reaction and so on [4–9]. During the process of deterioration, water directly participates in much physical deterioration and indirectly takes part in chemical deterioration as a medium of aggressive ions transport [8–11]. Therefore, the study of water transport in cement-based materials is the basis for their durability research, and it is of great significance to evaluate the properties, predict the service life and improve the design level of durability [10–14].

Water absorption is a reliable way of measuring the ability of a material to absorb and transmit water by capillarity [15]. A simple and widely used technique of water absorption is the gravimetric method, which was firstly used in porous material (soil) research in 1957 and 20 years later, it was used in the research of building materials, e.g. stone, brick, concrete, etc. [15–17]. From the gravimetric method, the rate of water absorption (sorptivity) can be obtained from the relationship

$$I = S \cdot t^{1/2}. \tag{1.1}$$

Where, $I$ is the volume of absorbed water per unit cross section (m), $S$ is the sorptivity (m/s$^{1/2}$) and $t$ is exposed time (s). With the gravimetric method, it is easy to obtain the amount of water absorption, however, it is impossible to observe the movement of water and determine the penetration depth.

The dynamic process of water transport in porous materials can be monitored by Neutron Radiography (NR) [18–21], Nuclear Magnetic Resonance (NMR) [22–25], Gamma ray [26] and X-ray Computed Tomography (X-ray CT) [11,27]. Based on these techniques, the distance of water uptake in material can be determined and the capillary coefficient can be calculated as follows:

$$x = k \cdot t^{1/2}, \tag{1.2}$$

where, $x$ is the distance of water uptake (m) and $k$ is the capillary coefficient (m/s$^{1/2}$).

According to the above introduction, $S$ describes the relationship between the amount of water absorption and exposed time while $k$ is the distance of water uptake in material as a function of exposed time. In general, $k$ is more favourable to be applied in the durability research of cement-based materials, since it directly describes the penetration depth at a given time. However, it is well known that $k$ is difficult to obtain unless special technologies (NR, NMR, Gamma ray or X-ray CT) are used while $S$ can be easily measured by the gravimetric method. Unfortunately, the relationship between $k$ and $S$ is not determined. Hanžič et al. [21] obtained the capillary coefficient of concrete using NR and calculated the ratio between $S$ and $k$, however, it is only an experiment result and a model was not provided, then the capillary coefficient cannot be calculated from the measurement of sorptivity if their ratio is not known. Therefore, it is important to establish the theoretical relationship between $k$ and $S$ and offer a reliable model.

In this work, the relationship between $k$ and $S$ is established based on theory analysis and experimental research. It aims to provide a simple method for the determination of capillary coefficient in accordance with the measurement of sorptivity.

## 2. Theory analysis

Water flow in porous media under pressure can be described as Darcy's Law [28,29]:

$$V_F = \frac{K}{\eta} \frac{\Delta P}{x}. \tag{2.1}$$

Where $V_F$ is the rate of water flux (m/s), $K$ is the intrinsic permeability (m$^2$), $\Delta P$ is the difference of pressure (Pa) and $\eta$ is the viscosity of liquid water (0.001 Pa·s, 20°C).

According to the Hagen–Poiseuille Law, the motion of water menisci in the pores [30]

$$K = \frac{r^2}{8}, \tag{2.2}$$

where $r$ is the radius of capillary pore (m).

For water absorption of capillary pore, the pressure gradient is contributed from the capillary force and described by Laplace equation [30]

$$\Delta P = \frac{2\gamma}{r} \cos \theta, \tag{2.3}$$

where $\gamma$ is the surface tension of water (0.072 N m$^{-1}$, 20°C) and $\theta$ is the contact angle.

For water flow through a capillary pore with the length of $x$ at any given time $t$, $V_F$ can also be described as follows:

$$V_F = \frac{dx}{dt}. \tag{2.4}$$

Combining equations (2.1)–(2.4) gives

$$x = \left(\frac{r\gamma}{2\eta}\cos\theta\right)^{1/2} t^{1/2} = k \cdot t^{1/2} \tag{2.5}$$

and

$$k = \left(\frac{r\gamma}{2\eta}\cos\theta\right)^{1/2}. \tag{2.6}$$

Equation (2.5) is the famous Lucas–Washburn equation, which describes the relationship between the distance of water flow and exposed time [31].

The volume of water flux can be described as

$$q = \pi r^2 \cdot x = \pi r^2 \left(\frac{r\gamma}{2\eta}\cos\theta\right)^{1/2} t^{1/2}, \tag{2.7}$$

where $q$ is the volume of water flow in capillary pore (m$^3$).

For water transport into a porous material in one dimension, all the pores in the cross section exposed to water can be equivalent to $N$ straight capillary pores with the radius of $r$. Then the volume of water absorption can be given by

$$Q = N \cdot q = N\pi r^2 \left(\frac{r\gamma}{2\eta}\cos\theta\right)^{1/2} t^{1/2}. \tag{2.8a}$$

And the mass of water absorption

$$M = \rho Q = N\pi r^2 \rho \left(\frac{r\gamma}{2\eta}\cos\theta\right)^{1/2} t^{1/2}, \tag{2.8b}$$

where $Q$ is the volume of absorbed water, $M$ is the mass of absorbed water and $\rho$ is the density of water (1000 kg m$^{-3}$, 20°C).

In accordance with the principle of stereology,

$$\frac{N \cdot \pi r^2}{A} = \varphi, \tag{2.9}$$

where $A$ is the section area of sample exposed to water and $\varphi$ is the porosity.

Then, equations (2.8a) and (2.8b) can be written as follows:

$$Q = A\varphi\left(\frac{r\gamma}{2\eta}\cos\theta\right)^{1/2} t^{1/2} \tag{2.10a}$$

and

$$M = A\varphi\rho\left(\frac{r\gamma}{2\eta}\cos\theta\right)^{1/2} t^{1/2}. \tag{2.10b}$$

If the water absorption of porous material is investigated by the gravimetric method, the result is usually described as [29]

$$I = \frac{M}{\rho A} = S \cdot t^{1/2}. \tag{2.11}$$

Combining equation (2.10b) and (2.11) gives

$$S = \varphi\left(\frac{r\gamma}{2\eta}\cos\theta\right)^{1/2}. \tag{2.12}$$

**Table 1.** Fundamental properties of cement.

| specific surface area (m$^2$/kg) | setting time (min) | | compressive strength (MPa) | | flexural strength (MPa) | |
|---|---|---|---|---|---|---|
| | initial | final | 3d | 28d | 3d | 28d |
| 375 | 150 | 204 | 35.2 | 66.4 | 6.2 | 9.3 |

**Table 2.** Chemical compositions of cement (wt%).

| composition | CaO | SiO$_2$ | Al$_2$O$_3$ | Fe$_2$O$_3$ | SO$_3$ | P$_2$O$_5$ | Na$_2$O | K$_2$O | others |
|---|---|---|---|---|---|---|---|---|---|
| cement | 55.02 | 20.32 | 7.83 | 2.79 | 4.74 | 5.23 | 0.33 | 0.42 | 3.32 |

**Table 3.** Mix proportions of mortar and concrete (wt%).

| sample | W/C | cement | sand | limestone | water |
|---|---|---|---|---|---|
| mortar | 0.45 | 100 | 164 | 0 | 45 |
| concrete | 0.45 | 100 | 151 | 258 | 45 |

Then, the relationship between capillary coefficient and sorptivity can be obtained from equations (2.6) and (2.12)

$$\frac{S}{k} = \varphi. \tag{2.13}$$

# 3. Experimental research

## 3.1. Raw materials

Chinese standard P·II 2.5 Portland cement was used in this work. Its fundamental properties and chemical compositions are shown in tables 1 and 2, respectively. Natural river sand with the fineness modulus of 2.96 was used and the apparent density is 2630 kg m$^{-3}$. Limestone with the particle size of 5–20 mm and apparent density of 2720 kg m$^{-3}$ was used for the preparation of concrete.

## 3.2. Sample preparation

Mortar and concrete with W/C = 0.45 were prepared in this work and their mix proportions are shown in table 3. The compressive strength of concrete at 28 days is 52 MPa. Cubic specimens with the size of 100 mm × 100 mm × 100 mm and cylindrical specimens with 100 mm in diameter and 200 mm in height were prepared. All the specimens were cured at 20 ± 1°C, RH ≥ 95% for 60 days. For the measurement of capillary coefficient using X-ray CT, samples with the size of 20 mm × 20 mm × 80 mm were cut from the middle of the cubic specimen. At the same time, samples of 100 mm in diameter and 10, 20, 30, 40 and 50 mm in thickness were cut from the cylindrical specimens, which were used for testing the property of the water absorption using the gravimetric method and measuring the porosity by the vacuum water-saturated method. All the samples were dried at 60°C in an oven for constant mass. For the test of water absorption, the surrounding surface and one section of the samples were covered using epoxy resin and the other section was used for exposing to liquid. For the measurement of porosity, three samples with 100 mm in diameter and 50 mm in thickness were used and not covered by epoxy resin.

## 3.3. Testing methods

### 3.3.1. X-CT combined with CsCl enhancing

A Y.CT Precision S X-ray CT scanner (made by YXLON, Germany) with 1024 detectors (Y.XRD 0820) was used in this work. The working voltage and current of X-ray tube were 195 kV and 0.34 mA, respectively.

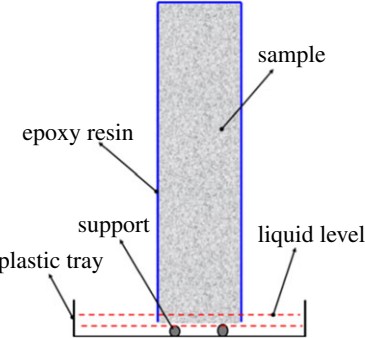

**Figure 1.** The placement of sample for CT test.

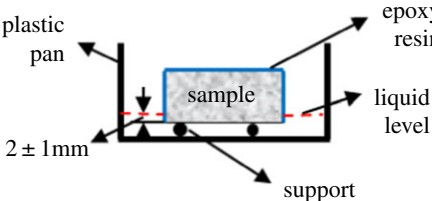

**Figure 2.** Schematic of water absorption.

**Table 4.** Surface tension of CsCl solution with different concentration (20°C).

| concentration (wt%) | 0 | 1 | 3 | 5 |
|---|---|---|---|---|
| surface tension ($10^{-3}$ N/m) | 72.26 | 71.55 | 72.09 | 72.38 |

For this test, a plastic tray with two supports at the bottom was prepared and the sample was fixed on the supports, as shown in figure 1. Next, put the tray on the sample platform of X-ray CT, added water solution with 5% (in weight) CsCl into the tray (the liquid level was 1–3 mm above the supports) and recorded the time once the liquid level contacted with the exposed surface of sample. The samples were scanned at fixed time and CT images were reconstructed by using VG Studio MAX software. This test was operated at room temperature (20 ± 1°C). CsCl was used due to three reasons [11]: (1) high atomic number of Cs (55), (2) the nature of Cs is similar to Na/K as they belong to the same main group in the periodic table of elements and (3) the addition of CsCl does not change the nature of liquid and the surface tension of CsCl solution with low concentration (less than or equal to 5%) is close to pure water (table 4).

### 3.3.2. Gravimetric method for water absorption

Water absorption measured by the gravimetric method was conducted in accordance with ASTM C1585-13 [30,32] and the schematic is shown in figure 2. The timing device was started when the sample contacted with water and the mass of sample was recorded at the time points of 5 min, 10 min, 20 min, 30 min, 60 min, every hour up to 6 h, 10 h, 14 h, 24 h and once a day up to 30 days. The temperature for this measurement of water absorption was controlled at 20 ± 1°C. The experiment result was described using equation (2.11).

### 3.3.3. Porosity measurement from vacuum water-saturated method

The porosity of mortar and concrete was measured using the vacuum water-saturated method. Three dried samples were saturated by water using a vacuum water-saturated machine, the procedure was conducted as follows: the samples were put in the container and the door screws were tightened then the container was vacuumized up to −98 kPa and held for 3 h; next, water was inhaled through a pipe until the samples were submerged; the container was vacuumized again up to −98 kPa and held

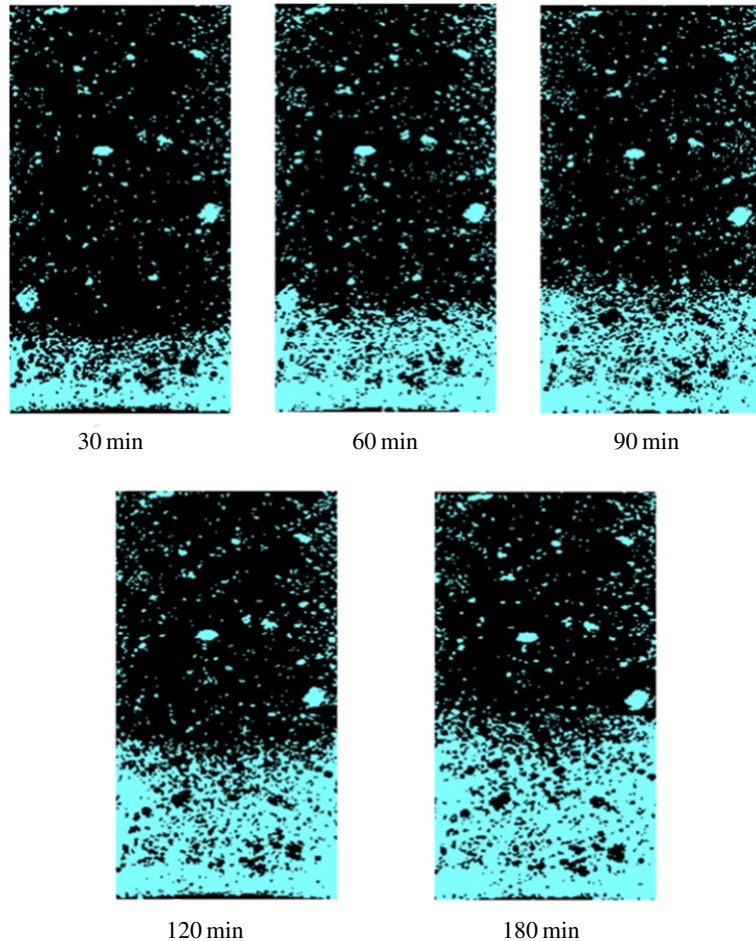

**Figure 3.** The evolution of water uptake in mortar.

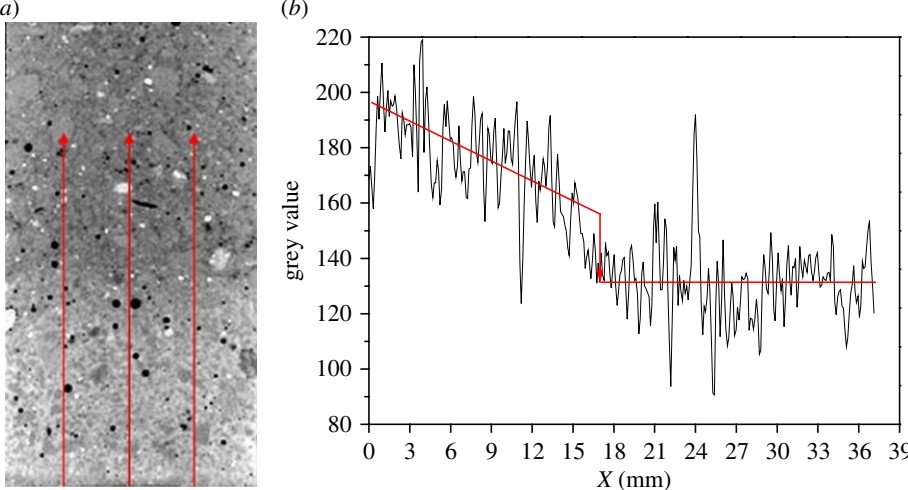

**Figure 4.** Determination of the height of water uptake from CT image. (*a*) CT image, (*b*) average grey value from three lines.

for 1 h; after that, it was quieted for 20 h. The whole process went ahead automatically based on fixed procedure. Then, the porosity of the sample can be calculated as follows:

$$\varphi = \frac{m_s - m_0}{\rho \cdot V_s}\,. \tag{3.1}$$

Where, $\varphi$ is the porosity of the sample, $m_s$ is the mass of sample at saturated state, $m_0$ is the mass of sample dried at 60°C for constant mass, $\rho$ is the density of water and $V_s$ is the volume of sample.

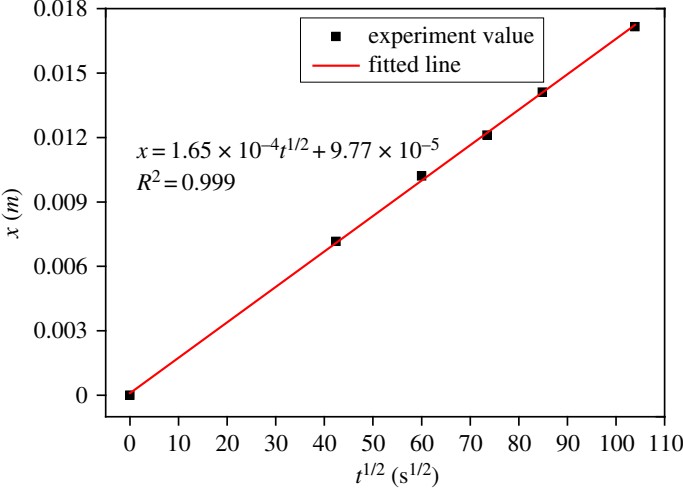

**Figure 5.** The distance of water uptake in mortar as a function of the square root of exposed time.

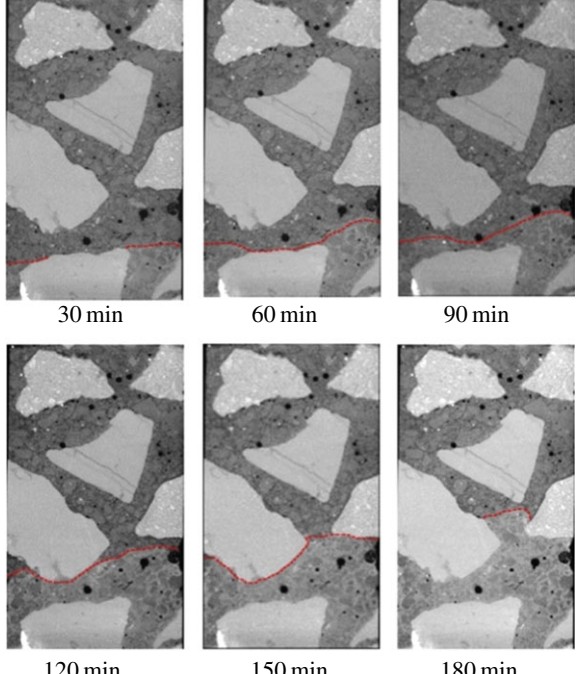

**Figure 6.** The evolution of water uptake in concrete.

# 4. Results and discussion

## 4.1. Dynamic process of water uptake in mortar and concrete monitored by X-ray CT

The evolution of water uptake in mortar was monitored *in situ* by the technique of X-ray CT combined with CsCl enhancing and the results are shown in figure 3. The bright zone of mortar indicates that this area has been filled by water. It can be seen that the height of water uptake in mortar increases gradually with the exposed time increasing. Figure 4 shows the method to determine the height of water uptake in mortar from the CT image. Three lines with an equal spacing were drawn on the CT image, as shown in figure 4*a*, then the grey value along the three lines was collected and the distribution of average grey value is shown in figure 4*b*. It is clear that the grey value has a sudden decrease at a certain distance and the height of water uptake in mortar can be determined from that.

In addition, figure 5 shows the relationship between the distance of water uptake ($x$) and exposed time ($t$), where $x$ is plotted against the square root of exposed time ($t^{1/2}$). According to the experiment results, a fitted line is obtained using the least square method, as follows:

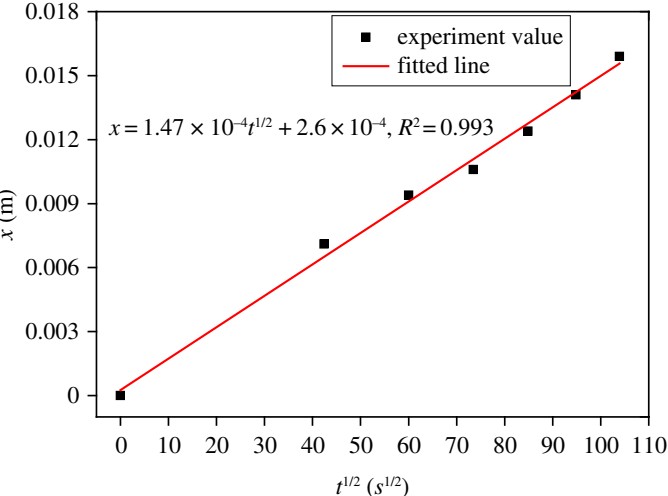

**Figure 7.** The distance of water uptake in concrete as a function of the square root of exposed time.

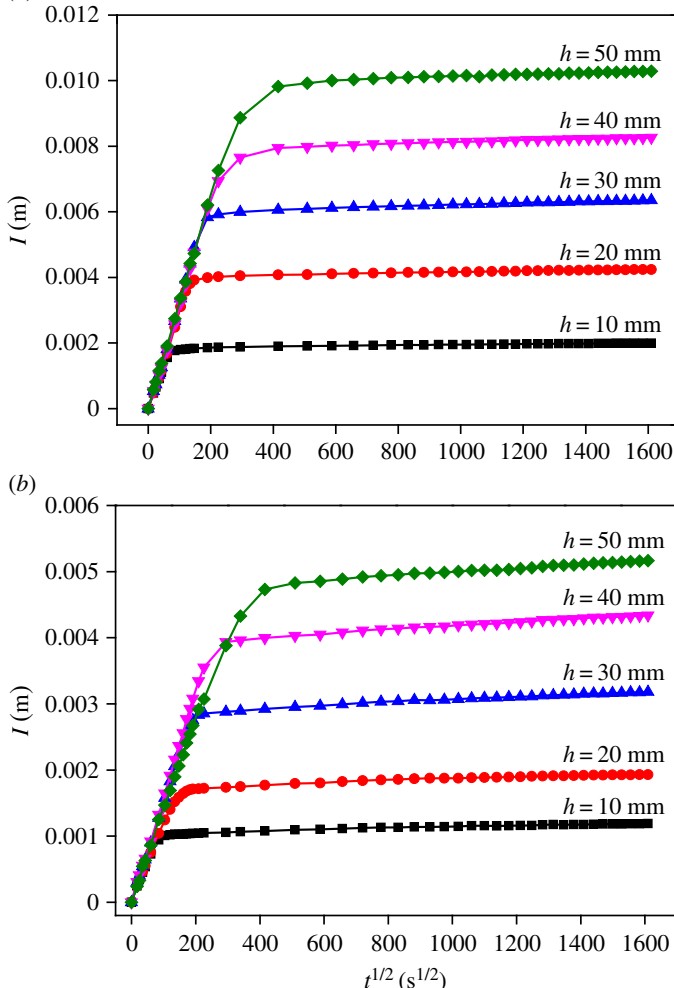

**Figure 8.** Cumulative water absorption of samples with different thickness. (*a*) Mortar and (*b*) concrete.

$$x = 1.65 \times 10^{-4} t^{1/2} + 9.77 \times 10^{-5}.$$

According to equation (2.5), the capillary coefficient of mortar equals the slope of the fitted line, that is, $k = 1.65 \times 10^{-4} \, \text{m/s}^{1/2}$, which describes the distance of water uptake in mortar as a function of exposed time.

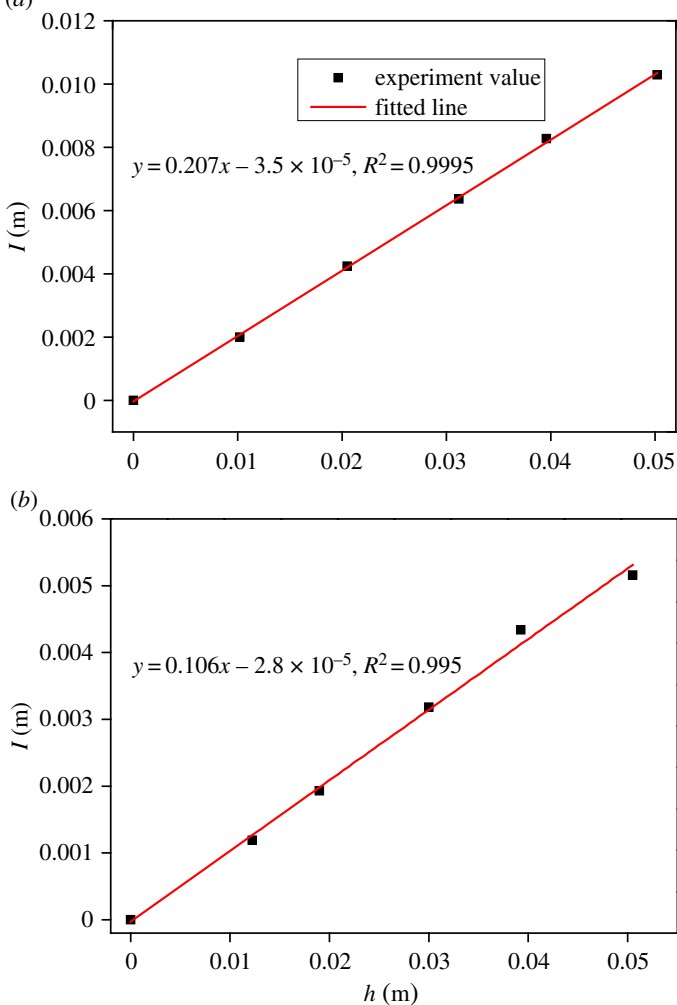

**Figure 9.** Relationship between total absorbed water for 30 days and the thickness of sample. (*a*) Mortar and (*b*) concrete.

Figure 6 shows the evolution of water uptake in concrete monitored by the technique of X-ray CT combined with CsCl enhancing. Water front bypasses coarse aggregate and goes up with the exposed time increasing. Figure 7 shows the distance of water uptake in concrete as a function of the square root of exposed time. Owing to the existence of coarse aggregate, the method to determine the distance of water uptake in mortar cannot be used in concrete, so the distance of water uptake in concrete is determined from the maximum height of water uptake. As shown in figure 7, the relationship between $x$ and $t^{1/2}$ is still linear for concrete and $k = 1.47 \times 10^{-4}$ m/s$^{1/2}$. It also can be seen that the capillary coefficient of concrete is lower than that of the mortar with same W/C due to the tortuosity effect of coarse aggregate.

## 4.2. Water absorption of mortar and concrete using the gravimetric method

The cumulative water absorption of mortar and concrete with different thicknesses ($h$) is shown in figure 8. According to the curves, the process of water absorption can be divided into two stages: rapid absorption and steady absorption. In the period of rapid absorption, the curves of water absorption are nearly coincident for the samples with different thickness; this stage is mainly controlled by capillary pores [33,34]. At the end of rapid absorption, water arrives at the top of samples through the capillary pores. After that, water continues to transport into gel pores, which is mainly controlled by the mechanism of diffusion [33,35].

Figure 9 shows the total amount of absorbed water for 30 days as a function of the thickness of the sample. Obviously, the relationship is linear and the slopes are 0.207 and 0.106 for mortar and concrete, respectively. That is, $\mathrm{d}(I)/\mathrm{d}(h) = 0.207$ for mortar and $\mathrm{d}(I)/\mathrm{d}(h) = 0.106$ for concrete. According to

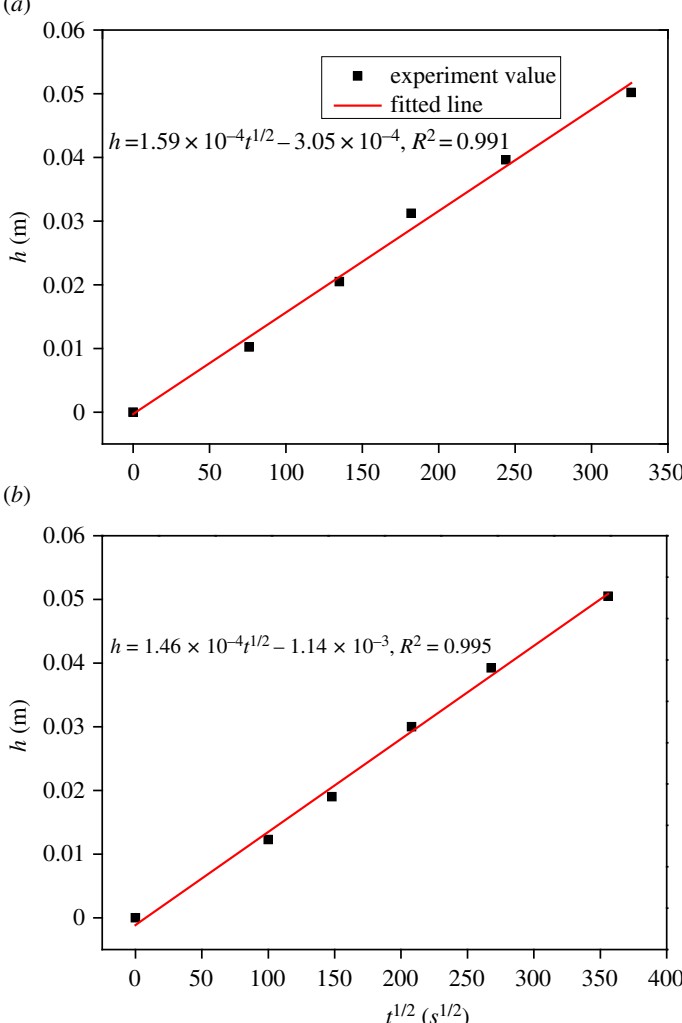

**Figure 10.** Relationship between the thickness of sample and the time water arrives at the top of sample. (*a*) Mortar and (*b*) concrete.

equation (2.11), the volume of water absorption can be given by

$$Q = \frac{M}{\rho} = I \cdot A. \tag{4.1}$$

The volume of the sample is described as

$$V = A \cdot h, \tag{4.2}$$

where $h$ is the thickness of the sample.

Then, the porosity of the sample can be calculated as follows:

$$\varphi = \frac{Q}{V} = \frac{I \cdot A}{A \cdot h} = \frac{I}{h}. \tag{4.3}$$

According to equation (4.3), it can be seen that the porosity of the sample equals the slope of the fitted line shown in figure 9. Then, the porosities of mortar and concrete are separately 0.207 and 0.106. In addition, the porosities of mortar and concrete measured by the vacuum water-saturated method are 0.217 and 0.113, respectively. The results obtained from two different methods are close to each other.

From figure 8, it can also be seen that the duration time of rapid absorption (i.e. the time water arrives at the top of sample) depends on the thickness of the sample. Figure 10 shows the relationship between the thickness of the sample and the time water arrives at the top of the sample. Clearly, the relation of $h$ and $t^{1/2}$ can be described as linear and the slopes of fitted lines are separately $1.59 \times 10^{-4}$ m/s$^{1/2}$ and $1.46 \times 10^{-4}$ m/s$^{1/2}$ for mortar and concrete, which can also be considered as the capillary coefficients

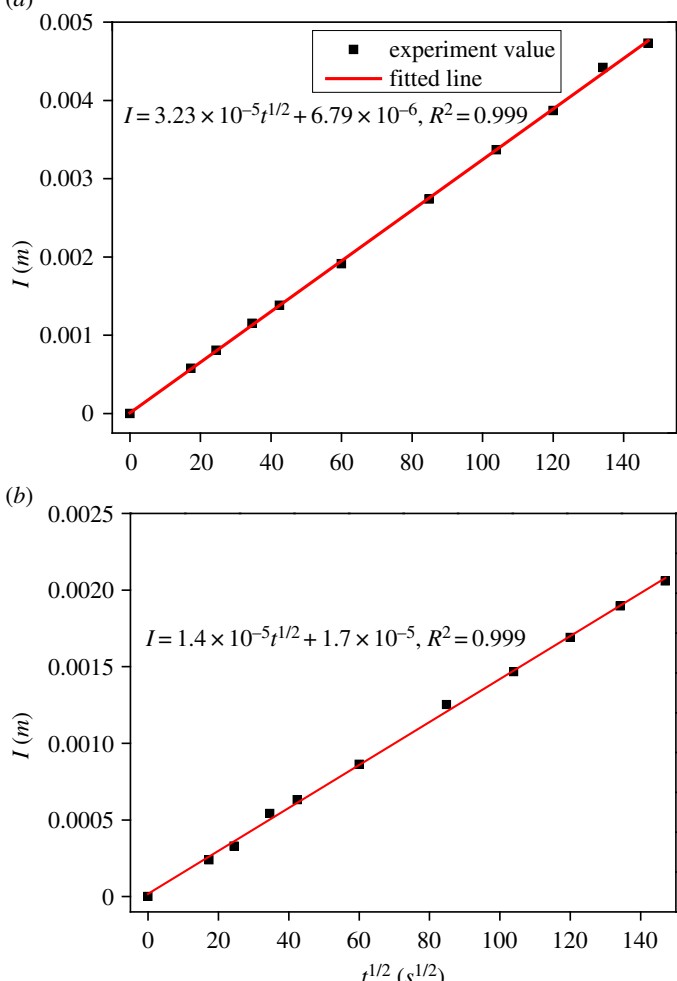

**Figure 11.** The amount of absorbed water as a function of the square root of exposed time. (*a*) Mortar and (*b*) concrete.

of mortar and concrete since they describe the distance of water uptake at certain exposed times. These capillary coefficients are close to that obtained from the technique of X-CT combined with CsCl enhancing.

In accordance with the plots of $I$ versus $t^{1/2}$ for mortar and concrete with the thickness of 50 mm, the sorptivities were obtained by fitting the data during the first 6 h, as shown in figure 11, which are $3.23 \times 10^{-5}$ m/s$^{1/2}$ and $1.4 \times 10^{-5}$ m/s$^{1/2}$ for mortar and concrete, respectively.

From the above experiment results, it can be obtained that $S/k \approx 0.2$ for mortar and 0.1 for concrete, which are close to the porosities of mortar and concrete, respectively. These results give reliable evidence to the theory analysis.

## 5. Conclusion

In this study, the relationship between the capillary coefficient and sorptivity was firstly established based on theory analysis. According to experiment results, the ratio of sorptivity to capillary coefficient was obtained, which agrees with the result of theory analysis and well verifies the theory model, $S/k = \varphi$. Based on this model, the capillary coefficient of cement-based materials can be determined in accordance with the sorptivity and porosity.

Data accessibility. This article does not contain any additional data.

Authors' contributions. L.Y. carried out the molecular laboratory work and data analysis and drafted the manuscript; D.G. participated in the design of the study and revised the manuscript; Y.Z. also participated in the design of the study and carried out data analysis; J.T. and Y.L. mainly participated in experiment work and data analysis. All authors gave final approval for publication and agreed to be held accountable for the work performed therein.

Competing interests. We have no competing interests.

Funding. This study was supported by National Natural Science Foundation of China (grant nos 51808508 and U1704254), China Postdoctoral Science Foundation (grant no. 2018M642786) and Jiangsu Key Laboratory for Construction Materials (grant no. CM2018-03).

Acknowledgements. We thank editors and anonymous reviewers for their helpful suggestions on the early versions of this manuscript.

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
