## [Reviewer comments · Royal Society Open Science]

Review History

RSOS-190112.R0 (Original submission)

Review form: Reviewer 1

Is the manuscript scientifically sound in its present form?

Yes

Are the interpretations and conclusions justified by the results?

Yes

Is the language acceptable?

Yes

Is it clear how to access all supporting data?

Yes

Do you have any ethical concerns with this paper?

No

Have you any concerns about statistical analyses in this paper?

No

Recommendation?

Accept with minor revision (please list in comments)

Comments to the Author(s)

Please see find the attached document (Appendix A).

Review form: Reviewer 2

Is the manuscript scientifically sound in its present form?

Yes

Are the interpretations and conclusions justified by the results?

Yes

Is the language acceptable?

Yes

Is it clear how to access all supporting data?

Yes

Do you have any ethical concerns with this paper?

No

Have you any concerns about statistical analyses in this paper?

No

Recommendation?

Accept with minor revision (please list in comments)

Comments to the Author(s)

In this work, X-ray CT is used to continuously monitor the dynamic process of water transport in mortar and concrete; meanwhile, the amount of water absorption is also measured by the gravimetric method. It is interesting that the relationship between the capillary coefficient and sorptivity is established based on theory analysis and experiment results. However, this paper should be revised before accepting. Some comments for revising are as follows:

(1) Two important references about the transport properties of cement-based materials are advised to cite in this paper.

Hou, D., Li, T., & Wang, P. (2018). ACS Sustainable Chemistry & Engineering, 6(7), 9498-9509.

Hou, D., Zhao, T., Ma, H., & Li, Z. (2015). The Journal of Physical Chemistry C, 119(3), 1346-1358.

(2) Page 16: Table 3, the units are not given.

(3) The porosity of sample measured by the method of water absorption only includes open pores, how about the effect of closed pores on the water absorption?

(4) In this model, $S/k = \phi$, the porosity (ϕ) is measured by vacuum water-saturated method, why? Can it be measured by others (e.g., MIP)? This should be paid more attention.

Review form: Reviewer 3

Is the manuscript scientifically sound in its present form?

Yes

Are the interpretations and conclusions justified by the results?

Yes

Is the language acceptable?

Yes

Is it clear how to access all supporting data?

No

Do you have any ethical concerns with this paper?

No

Have you any concerns about statistical analyses in this paper?

No

Recommendation?

Accept with minor revision (please list in comments)

Comments to the Author(s)

The durability of cement-based materials is a critical property during their total service life and water absorption can directly reflect it. Then, the researcher and engineer usually measure the water absorption to predict their long time performance. In this work, X-ray CT and the gravimetric method were used together to describe water absorption of mortar and concrete; what's more, the relationship between the capillary coefficient and sorptivity was established based on theory analysis and experiment. It offers a model for the determination of capillary coefficient in accordance with the measurement of sorptivity. This paper can be accepted in this journal after a minor revision. However, some improvements should be done:

1. Water absorption was measured by X-ray CT and the gravimetric method in this work, if the temperature for these tests are the same?
2. The hydration of cement is a long time process, if the second hydration is considered during the process of water absorption?
3. In Fig.3, some bright spots that appear above the fluid penetration area, what are they?
4. The reviewer think that only capillary pores are considered in this work, how about gel pores in C-S-H?
5. Please check the language again.

Review form: Reviewer 4 (Hongjian Du)

Is the manuscript scientifically sound in its present form?

Yes

Are the interpretations and conclusions justified by the results?

Yes

Is the language acceptable?

Yes

Is it clear how to access all supporting data?

Yes

Do you have any ethical concerns with this paper?

No

Have you any concerns about statistical analyses in this paper?

No

Recommendation?

Accept with minor revision (please list in comments)

Comments to the Author(s)

This is a very interesting work to propose a simple model to quantify the relationship between porosity, sorptivity and capillary coefficient. Both theoretical and experimental works are conducted to support the hypothesis. Overall, this paper is well written and can be accepted after revising the following:

Abstract: It is debatable to state that all durability performances of cement-based materials depend on water absorption. For example, chloride can diffuse into concrete, which is governed, by both concrete quality and concentration difference.

Decision letter (RSOS-190112.R0)

16-Apr-2019

Dear Dr Yang,

The editors assigned to your paper ("Relationship between sorptivity and capillary coefficient for water absorption of cement-based materials: theory analysis and experiment") have now received comments from reviewers. We would like you to revise your paper in accordance with the referee and Associate Editor suggestions which can be found below (not including confidential reports to the Editor). Please note this decision does not guarantee eventual acceptance.

Please submit a copy of your revised paper before 09-May-2019. Please note that the revision deadline will expire at 00.00am on this date. If we do not hear from you within this time then it will be assumed that the paper has been withdrawn. In exceptional circumstances, extensions may be possible if agreed with the Editorial Office in advance. We do not allow multiple rounds of revision so we urge you to make every effort to fully address all of the comments at this stage. If deemed necessary by the Editors, your manuscript will be sent back to one or more of the original reviewers for assessment. If the original reviewers are not available, we may invite new reviewers.

- Data accessibility

If you wish to submit your supporting data or code to Dryad (<http://datadryad.org/>), or modify your current submission to dryad, please use the following link:
<http://datadryad.org/submit?journalID=RSOS&manu=RSOS-190112>

- Competing interests

- Authors' contributions

- Acknowledgements

- Funding statement

on behalf of Professor R. Kerry Rowe (Subject Editor)
 openscience@royalsociety.org

Associate Editor's comments:

Please fully incorporate the scientific changes required by the reviewers in your revision. Furthermore, you should seek the advice of a language polishing service to improve the language quality in the manuscript. You may find examples of such services at <https://royalsociety.org/journals/authors/language-polishing/>.

Comments to Author:

Reviewers' Comments to Author:

Reviewer: 1

Comments to the Author(s)

Please see find the attached document.

Reviewer: 2

Comments to the Author(s)

In this work, X-ray CT is used to continuously monitor the dynamic process of water transport in mortar and concrete; meanwhile, the amount of water absorption is also measured by the gravimetric method. It is interesting that the relationship between the capillary coefficient and sorptivity is established based on theory analysis and experiment results. However, this paper should be revised before accepting. Some comments for revising are as follows:

(1) Two important references about the transport properties of cement-based materials are advised to cite in this paper.

Hou, D., Li, T., & Wang, P. (2018). *ACS Sustainable Chemistry & Engineering*, 6(7), 9498-9509.

Hou, D., Zhao, T., Ma, H., & Li, Z. (2015). *The Journal of Physical Chemistry C*, 119(3), 1346-1358.

(2) Page 16: Table 3, the units are not given.

(3) The porosity of sample measured by the method of water absorption only includes open pores, how about the effect of closed pores on the water absorption?

(4) In this model, $S/k = \phi$, the porosity (ϕ) is measured by vacuum water-saturated method, why? Can it be measured by others (e.g., MIP)? This should be paid more attention.

Reviewer: 3

Comments to the Author(s)

The durability of cement-based materials is a critical property during their total service life and water absorption can directly reflect it. Then, the researcher and engineer usually measure the water absorption to predict their long time performance. In this work, X-ray CT and the gravimetric method were used together to describe water absorption of mortar and concrete; what's more, the relationship between the capillary coefficient and sorptivity was established based on theory analysis and experiment. It offers a model for the determination of capillary coefficient in accordance with the measurement of sorptivity. This paper can be accepted in this journal after a minor revision. However, some improvements should be done:

1. Water absorption was measured by X-ray CT and the gravimetric method in this work, if the temperature for these tests are the same?
2. The hydration of cement is a long time process, if the second hydration is considered during the process of water absorption?
3. In Fig.3, some bright spots that appear above the fluid penetration area, what are they?
4. The reviewer think that only capillary pores are considered in this work, how about gel pores in C-S-H?
5. Please check the language again.

Reviewer: 4

Comments to the Author(s)

This is a very interesting work to propose a simple model to quantify the relationship between porosity, sorptivity and capillary coefficient. Both theoretical and experimental works are conducted to support the hypothesis. Overall, this paper is well written and can be accepted after revising the following:

Abstract: It is debatable to state that all durability performances of cement-based materials depend on water absorption. For example, chloride can diffuse into concrete, which is governed, by both concrete quality and concentration difference.

Author's Response to Decision Letter for (RSOS-190112.R0)

See Appendix B.

RSOS-190112.R1 (Revision)

Review form: Reviewer 2

Is the manuscript scientifically sound in its present form?

Yes

Are the interpretations and conclusions justified by the results?

Yes

Is the language acceptable?

Yes

Is it clear how to access all supporting data?

Yes

Do you have any ethical concerns with this paper?

No

Have you any concerns about statistical analyses in this paper?

No

Recommendation?

Accept as is

Comments to the Author(s)

The authors have addressed all the issues and the revised manuscript can be accept

Review form: Reviewer 3

Is the manuscript scientifically sound in its present form?

Yes

Are the interpretations and conclusions justified by the results?

Yes

Is the language acceptable?

Yes

Is it clear how to access all supporting data?

Not Applicable

Do you have any ethical concerns with this paper?

No

Have you any concerns about statistical analyses in this paper?

I do not feel qualified to assess the statistics

Recommendation?

Accept as is

Comments to the Author(s)

Authors had responded adequately

Decision letter (RSOS-190112.R1)

17-May-2019

Dear Dr Yang:

On behalf of the Editors, I am pleased to inform you that your Manuscript RSOS-190112.R1 entitled "Relationship between sorptivity and capillary coefficient for water absorption of cement-based materials: theory analysis and experiment" has been accepted for publication in Royal Society Open Science subject to minor revision in accordance with the referee suggestions. Please find the referees' comments at the end of this email.

The reviewers and Subject Editor have recommended publication, but also suggest some minor revisions to your manuscript. Therefore, I invite you to respond to the comments and revise your manuscript.

- Ethics statement

- Data accessibility

If you wish to submit your supporting data or code to Dryad (<http://datadryad.org/>), or modify your current submission to dryad, please use the following link:
<http://datadryad.org/submit?journalID=RSOS&manu=RSOS-190112.R1>

- Competing interests

- Authors' contributions

- Acknowledgements

- Funding statement

Because the schedule for publication is very tight, it is a condition of publication that you submit the revised version of your manuscript before 26-May-2019. Please note that the revision deadline will expire at 00.00am on this date. If you do not think you will be able to meet this date please let me know immediately.

Supplementary files will be published alongside the paper on the journal website and posted on

the online figshare repository (<https://figshare.com>). The heading and legend provided for each supplementary file during the submission process will be used to create the figshare page, so please ensure these are accurate and informative so that your files can be found in searches. Files on figshare will be made available approximately one week before the accompanying article so that the supplementary material can be attributed a unique DOI.

Kind regards,
Alice Power
Royal Society Open Science
openscience@royalsociety.org

on behalf of R. Kerry Rowe (Subject Editor)
openscience@royalsociety.org

Editorial Office Comments to Author:

You indicate that specialist language editing has been conducted on your work, and the Editors are grateful for this; however, you have provided no evidence to show what specialist was consulted nor whether the changes were incorporated. Please provide both.

Reviewer comments to Author:
Reviewer: 3

Comments to the Author(s)
Authors had responded adequately

Reviewer: 2

Comments to the Author(s)
The authors have addressed all the issues and the revised manuscript can be accept

Author's Response to Decision Letter for (RSOS-190112.R1)

See Appendix C.

Decision letter (RSOS-190112.R2)

31-May-2019

Dear Dr Yang,

I am pleased to inform you that your manuscript entitled "Relationship between sorptivity and

capillary coefficient for water absorption of cement-based materials: theory analysis and experiment" is now accepted for publication in Royal Society Open Science.

on behalf of Prof R. Kerry Rowe (Subject Editor)
openscience@royalsociety.org

Follow Royal Society Publishing on Twitter: [@RSocPublishing](https://twitter.com/RSocPublishing)
Follow Royal Society Publishing on Facebook:
<https://www.facebook.com/RoyalSocietyPublishing.FanPage/>
Read Royal Society Publishing's blog: <https://blogs.royalsociety.org/publishing/>

Appendix A

Water absorption is an important property to the cement-based materials, which is directly related to their durability. The amount of absorbed water and transport depth are usually used to describe the property of water absorption, however, their relationship is not clear. This work offers a technique of X-ray CT combined with CsCl enhancing to continuously monitor the dynamic process of water uptake in cement-based materials and measures the amount of water absorption by the gravimetric method, in further, the relationship between the capillary coefficient and sorptivity is established based on theory analysis and experiment results. It is an interesting and significant research work in this area, which can be published in this journal after a minor revision. Some suggestions are as follows:

- (1) Why the total amount of chemical compositions in cement is not 100%?
- (2) The unit for mix proportions of mortar and concrete shown in Table 3 is not given.
- (3) “.” should be added after “in this work”.
- (4) The porosity of sample was measured using vacuum water-saturated method, why not MIP? In addition, the porosities of mortar and concrete measured by vacuum water-saturated method are higher than that calculated from equation (19), they are different from each other.

Appendix B

Dear editor,

Thank you very much for your email dated on Apr. 16, 2019.

I carefully revised the manuscript in accordance with editor's and reviewers' comments, which are also responded as follows:

Associate Editor's comments:

Please fully incorporate the scientific changes required by the reviewers in your revision. Furthermore, you should seek the advice of a language polishing service to improve the language quality in the manuscript. You may find examples of such services at <https://royalsociety.org/journals/authors/language-polishing/>.

Response: Thank the editor very much! The authors have revised the manuscript according to the reviewers' comments. In addition, the language quality in the manuscript has been improved with the help of specialist.

Reviewers' Comments to Author:

Reviewer: 1

Water absorption is an important property to the cement-based materials, which is directly related to their durability. The amount of absorbed water and transport depth are usually used to describe the property of water absorption, however, their relationship is not clear. This work offers a technique of X-ray CT combined with CsCl enhancing to continuously monitor the dynamic process of water uptake in cement-based materials and measures the amount of water absorption by the gravimetric method, in further, the relationship between the capillary coefficient and sorptivity is established based on theory analysis and experiment results. It is an interesting and significant research work in this area, which can be published in this journal after a minor revision. Some suggestions are as follows:

(1) Why the total amount of chemical compositions in cement is not 100%?

Response: More chemical compositions with lower content were not shown in the table. Table 2 has been revised.

(2) The unit for mix proportions of mortar and concrete shown in Table 3 is not given.

Response: The unit "(wt%)" is added in Table 3.

(3) "." should be added after "in this work".

Response: "." has been added in the text.

(4) The porosity of sample was measured using vacuum water-saturated method, why not MIP? In addition, the porosities of mortar and concrete measured by vacuum water-saturated method are higher than that calculated from equation (19), they are different from each other.

Response: Vacuum water-saturated is a simple method to measure the porosity of porous materials, which can be operated easily. However, MIP is a special method to obtain the porosity of material, which is expensive and harmful to people. What's more, an applied pressure (>200 MPa) is necessary for MIP, which certainly destroy actual pore structure of material. Then, the porosity measured by MIP is not the actual porosity of material.

Although the porosities of mortar and concrete measured by vacuum water-saturated method are higher than that calculated from equation (19), the maximum difference is merely 6.6%. Then, the authors describe that “the results obtained from two different methods are close to each other.”

Reviewer: 2

In this work, X-ray CT is used to continuously monitor the dynamic process of water transport in mortar and concrete; meanwhile, the amount of water absorption is also measured by the gravimetric method. It is interesting that the relationship between the capillary coefficient and sorptivity is established based on theory analysis and experiment results. However, this paper should be revised before accepting. Some comments for revising are as follows:

(1) Two important references about the transport properties of cement-based materials are advised to cite in this paper.

Hou, D., Li, T., & Wang, P. (2018). ACS Sustainable Chemistry & Engineering, 6(7), 9498-9509.

Hou, D., Zhao, T., Ma, H., & Li, Z. (2015). The Journal of Physical Chemistry C, 119(3), 1346-1358.

Response: These two references have been cited in this paper.

(2) Page 16: Table 3, the units are not given.

Response: The unit “(wt%)” is added in Table 3.

(3) The porosity of sample measured by the method of water absorption only includes open pores, how about the effect of closed pores on the water absorption?

Response: In fact, water absorption and ions transport are mainly affected by the open pores of material, and then the effect of closed pores on water absorption is not considered.

(4) In this model, $S/k=\phi$, the porosity (ϕ) is measured by vacuum water-saturated method, why? Can it be measured by others (e.g., MIP)? This should be paid more attention.

Response: It is easy to measure the porosity of porous materials using vacuum water-saturated method, besides, which is the actual porosity. Usually, the porosity also can be measured by MIP, nitrogen adsorption, imaging method and so on.

However, these methods have their own shortcoming. MIP is a special method to obtain the porosity of material, which is expensive and harmful to people. What's more, an applied pressure (>200 MPa) is necessary for MIP, which certainly destroy actual pore structure of material. Then, the porosity measured by MIP is not the actual porosity of material. Nitrogen adsorption is not suitable for the measurement of pore with the diameter >50 nm. The imaging method cannot be used to characterize the micro-scale pores due to its lower resolution of image. Then, the porosity measured by nitrogen adsorption and imaging method is lower than the actual porosity of material. In this model, $S/k=\phi$, the porosity (ϕ) is measured by vacuum water-saturated method.

Reviewer: 3

The durability of cement-based materials is a critical property during their total service life and water absorption can directly reflect it. Then, the researcher and engineer usually measure the water absorption to predict their long time performance. In this work, X-ray CT and the gravimetric method were used together to describe water absorption of mortar and concrete; what's more, the relationship between the capillary coefficient and sorptivity was established based on theory analysis and experiment. It offers a model for the determination of capillary coefficient in accordance with the measurement of sorptivity. This paper can be accepted in this journal after a minor revision. However, some improvements should be done:

1. Water absorption was measured by X-ray CT and the gravimetric method in this work, if the temperature for these tests are the same?

Response: The temperature for water absorption measured by X-ray CT and the gravimetric method was controlled at $20\pm 1^\circ\text{C}$.

2. The hydration of cement is a long time process, if the second hydration is considered during the process of water absorption?

Response: In this work, the curing time of sample was increased up to 60 days to reduce the effect of second hydration on water absorption.

3. In Fig.3, some bright spots that appear above the fluid penetration area, what are they?

Response: The bright spots that appear above the fluid penetration are the sand with bigger size, whose density is higher than the paste, and then they are bright in the image.

4. The reviewer think that only capillary pores are considered in this work, how about gel pores in C-S-H?

Response: The process of water absorption can be divided into two stages: rapid absorption and steady absorption. In the period of rapid absorption, water uptake in

sample through the capillary pores and this stage is mainly controlled by capillary pores. In the period of steady absorption, water continues to transport into gel pores and it is mainly controlled by the mechanism of diffusion. Gel pores in C-S-H is less than 10 nm and the volume fraction is usually less than 10%, then the effect of gel pores on water absorption is usually not considered.

5. Please check the language again.

Response: The authors have revised the manuscript with the help of specialist.

Finally, we thank the reviewers very much for giving us such good suggestions to improve our manuscript. We also thank the editor very much for your time and effort to publish this paper.

Sincerely yours,

Lin Yang

Appendix C

Dear editor,

Thank you very much for your email dated on May. 17, 2019.

The comments are also responded as follows:

Editorial Office Comments to Author:

You indicate that specialist language editing has been conducted on your work, and the Editors are grateful for this; however, you have provided no evidence to show what specialist was consulted nor whether the changes were incorporated. Please provide both.

Response: Thank the editor again! The last revised manuscript was checked by Max Bates, an English native speaker, who works in Swinburne University of Technology. Besides, the changes were marked as red in the revised manuscript.

Reviewer comments to Author:

Reviewer: 3

Comments to the Author(s)

Authors had responded adequately

Response: Thank the reviewer very much!

Reviewer: 2

Comments to the Author(s)

The authors have addressed all the issues and the revised manuscript can be accept

Response: Thank the reviewer very much!

Sincerely yours,

Lin Yang